# Premature vision drives aberrant development of response properties in primary visual cortex

Sophie V Griswold[1,2], Stephen D Van Hooser[1,2,3]*

[1]Department of Biology, Brandeis University, Waltham, United States; [2]Volen Center for Complex Systems, Brandeis University, Waltham, United States; [3]Sloan-Swartz Center for Theoretical Neurobiology, Brandeis University, Waltham, United States

## eLife Assessment

This carefully conducted study aims to understand how the early visual experience of premature infants induces lasting deficits, including compromised motion processing. The authors address this **important** question in a ferret animal model, exposing the developing visual system prematurely to patterned visual input by opening one or both eyes at a time when both retinal waves and light traveling through closed lids can drive sensory responses. **Convincing** evidence is presented, suggesting that eye opening at this time impacts temporal frequency tuning and elevates spontaneous firing rates. These findings will have great relevance for neuroscientists studying visual system development, particularly in the context of premature birth.

*For correspondence:
vanhoosr@brandeis.edu

Competing interest: The authors declare that no competing interests exist.

**Abstract** Development of the mammalian visual system is thought to proceed in two stages. In the first stage, before birth in primates and before eye opening in altricial mammals, spontaneous activity generated by the retina and cortex shapes visual brain circuits in an activity-dependent but experience-independent manner. In the second stage, visual activity generated by sensory experience refines receptive fields. Here, we investigated the consequences of altering this sequence of events by prematurely opening one or both eyes of ferrets and examining visual receptive fields in monocular cortex after the closure of the critical period for ocular dominance plasticity. We observed that many cells in animals with prematurely opened eyes exhibited low-pass temporal frequency tuning and increased temporal frequency bandwidths, and these cells showed slightly increased orientation and direction selectivity index values. Spontaneous activity was greatly elevated in both hemispheres following the premature opening of one or both eyes, suggesting a global change in circuit excitability that was not restricted to cells that viewed the world through the prematurely opened eye. No major changes were noted in spatial frequency tuning. These results suggest that premature visual experience alters circuit excitability and visual receptive fields, in particular with respect to temporal processing. We speculate that closed lids in altricial mammals serve to prevent visual experience until circuits are initially established and are ready to be refined by visual experience.

## Introduction

The proper development of visual cortical circuits depends on both experience-independent (*Katz and Shatz, 1996*; *Wong, 1999*) and experience-dependent (*Huberman et al., 2008*; *Wang et al., 2010*; *Hensch and Quinlan, 2018*) processes. Classically, these mechanisms have been divided into two sequential stages: an early stage, before the onset of visual experience, in which molecular cues

**eLife digest** The developing brain requires the right experiences at the right time to form proper connections. For most mammals, vision develops in two main stages. Before birth in humans – or before eye opening in animals born with closed eyes – spontaneous activity within the brain helps establish an initial visual circuitry without external visual input. These initial connections are then fine-tuned after birth or eye opening.

This sequential development is critical for normal visual function. Moreover, the visual system processes different aspects of vision (like motion, direction and detail) through specialized neural pathways, and proper development of these pathways also requires the right sequence of experiences.

In humans, babies born prematurely receive visual stimulation during what would typically be a pre-visual period, potentially disrupting this sequence. To find out what happens when visual experience occurs too early Griswold and Van Hooser studied the developing visual system in ferrets by artificially opening one or both eyes before they would naturally open and recording how neurons of the visual system responded to moving stimuli.

The results showed that ferrets with prematurely opened eyes developed significant abnormalities in their visual processing, particularly in how neurons respond to moving stimuli. Using electrophysiological recordings in the primary visual cortex, Griswold and Van Hooser observed that neurons in the ferrets showed unusually broad responses to different speeds of motion and increased activity at slow speeds. These neurons also showed higher spontaneous activity levels and increased suppression of activity in response to certain stimuli, such as quickly moving grating stimuli. These changes in firing rates and suppression occurred even in a part of the visual cortex that was ipsilateral (on the same side) to the early-opened eye, a location that does not receive direct input from the early-opened eye, suggesting that premature vision causes widespread alterations in brain circuit development. Other aspects of vision, like the ability to detect detail, were less affected.

These findings suggest that the timing of visual experiences is critical for healthy development and may help explain why human infants born very prematurely often have difficulty with motion perception, even without brain injuries. Understanding these neural differences could guide the development of targeted visual therapies for premature infants. Further research could determine if specific types of visual stimulation (or a lack of stimulation) might be protective during this sensitive period.

and spontaneous activity in retina and cortex guide initial circuit formation (*Katz and Shatz, 1996*; *Wong, 1999*; *Grubb et al., 2003*; *Cang et al., 2005*; *Smith et al., 2018*), followed by a period in which activity driven by visual experience refines circuit function (*Chapman et al., 1996*; *Huberman et al., 2008*; *Li et al., 2006*; *Wang et al., 2010*; *Chang et al., 2020*). The sequential nature of these two phases presumably ensures a developmental trajectory in which circuits achieve a state of maturation that provides proper support for the onset of experience-driven mechanisms that finalize the construction process.

Given the sequential nature of visual development, it is interesting to ask if there are any positive or negative consequences of violating this sequence by introducing patterned visual stimulation at earlier time points. Are there significant changes that cannot be reversed even with subsequent visual experience? These questions are especially important to address in light of the rising population of human infants born very prematurely (*Crump et al., 2019*), who receive visual stimulation during what is typically a presensory period (*Colonnese et al., 2010*; *Murata and Colonnese, 2016*).

To address these questions, we have examined the impact of premature visual experience on receptive fields in visual cortex of the ferret. Ferrets undergo a long postnatal developmental period and have a history of foundational developmental studies in vision. Ferrets are born with their eyes closed – a condition that is maintained for 30–35 days (*Issa et al., 1999*) – making it possible to introduce inappropriately early visual experience by opening the lids with forceps, without introducing factors related to immature birth. We opened zero, one, or both eyes early and evaluated receptive fields in primary visual cortex electrophysiologically after the closure of the critical periods for direction selectivity and ocular dominance. We studied monocular receptive fields only, so that we could be sure of the developmental condition of the eye, necessarily the contralateral eye, that was driving the receptive field properties we observed.

We observed substantial changes in temporal frequency tuning in monocular cells that observed the world through an early-opened eye. These cells, on average, exhibited strikingly broad tuning for temporal frequency with markedly enhanced responses at low temporal frequencies. These major alterations in visual selectivity were not present in all receptive field properties, as orientation and direction tuning in these cells were only slightly altered. By contrast, we observed substantial changes in spontaneous firing rates in both hemispheres after either or both eyes were opened early, suggesting that premature vision has a more global impact on the establishment of a firing rate set point in cortex (*Turrigiano and Nelson, 2004*; *Hengen et al., 2013*).

In all, these results suggest that premature visual experience causes lasting changes in temporal processing and in the baseline tone of cortical circuits. The changes in temporal frequency tuning are likely to have an impact on motion perception. To perceive a moving object, an animal must have knowledge of the object's direction and speed. The ratio of the temporal frequency of a stimulus to its spatial frequency is its speed, so temporal frequency tuning provides critical information about stimulus speed. Human babies that are born very prematurely are a heterogeneous group with a wide variety of visual deficits (*Kozeis, 2010*; *Dutton, 2013*; *Sakki et al., 2018*), but even when one excludes cases of known brain or retinal damage, very premature babies exhibit increased thresholds for motion perception later in life (*Taylor et al., 2009*; *Hou et al., 2011*). Our results in ferrets raise the possibility that inappropriately early visual experience could be an important factor that contributes to lasting motion processing deficits in humans born very prematurely.

Part of this work appeared as the PhD thesis of SVG (*Griswold, 2024*).

## Results

Our primary goal was to explore the influence of premature vision on the parameters of receptive fields in V1. We reared ferrets under three conditions in order to examine the impact of premature vision through one or both eyes in comparison with controls (*Figure 1*): in one group, we opened one eye early; in a second group, we opened both eyes early; and a third, control group was allowed to open their eyes naturally. Animals in both of the premature eye opening groups experienced early vision beginning on postnatal day 25, a developmental timepoint when retinal waves are still occurring (*Meister et al., 1991*; *Wong et al., 1993*), yet V1 is also responsive to visual input through the closed lids (*Krug et al., 2001*; *Akerman et al., 2002*). This ensured interaction between spontaneous, endogenously generated activity and patterned visual input. We hypothesized that this interaction may drive aberrant development of cortical receptive fields, as it is known that visual input is capable of initiating retinal waves through the closed lids (*Tiriac et al., 2018*). Following premature eye opening, animals were brought to the lab for 2 hr a day of unguided visual exposure over 4 days, for a total of 8 hr of visual exposure. Visual exposure consisted of gentle handling to prevent the young kits from sleeping, thus ensuring a sufficient amount of visual experience through the prematurely opened eye(s). Animals received no other intervention between the end of visual stimulus training on P28 and terminal electrophysiology experiments which occurred between P55-68.

We made multichannel recordings at a developmental time point when past studies have shown that V1 receptive fields are largely mature. By postnatal day 42, contrast sensitivity (*Li et al., 2006*), orientation (*Chapman and Stryker, 1993*), direction, temporal frequency, and spatial frequency (*Li et al., 2006*) tuning have reached adult-like levels and ocular dominance plasticity is greatly reduced (*Issa et al., 1999*). Thus, if premature vision causes permanent changes to receptive field properties, we ought to be able to see them at P55-68.

We recorded responses to long batteries of visual stimulation with sinusoidal gratings where multiple parameters were co-varied in order to assess a wide variety of possible receptive field differences between control animals and animals that had their eyes opened prematurely. In this study, all recordings were performed in the monocular visual cortex in order to simplify the interpretation of these changes without needing to consider interactions between the two eyes. We labeled cells as EO1contra (monocular neurons contralateral to a single early-opened eye), EO1ipsi (monocular neurons ipsilateral to a single early-opened eye), EO2 (monocular neurons contralateral to an early opened eye where both eyes were opened early), and control (monocular neurons in control animals). EO1 contra and EO2 cells observed the world through early-opened eyes, while EO1ipsi neurons observed the world through eyes that opened on time but were ipsilateral to early-opened eyes.

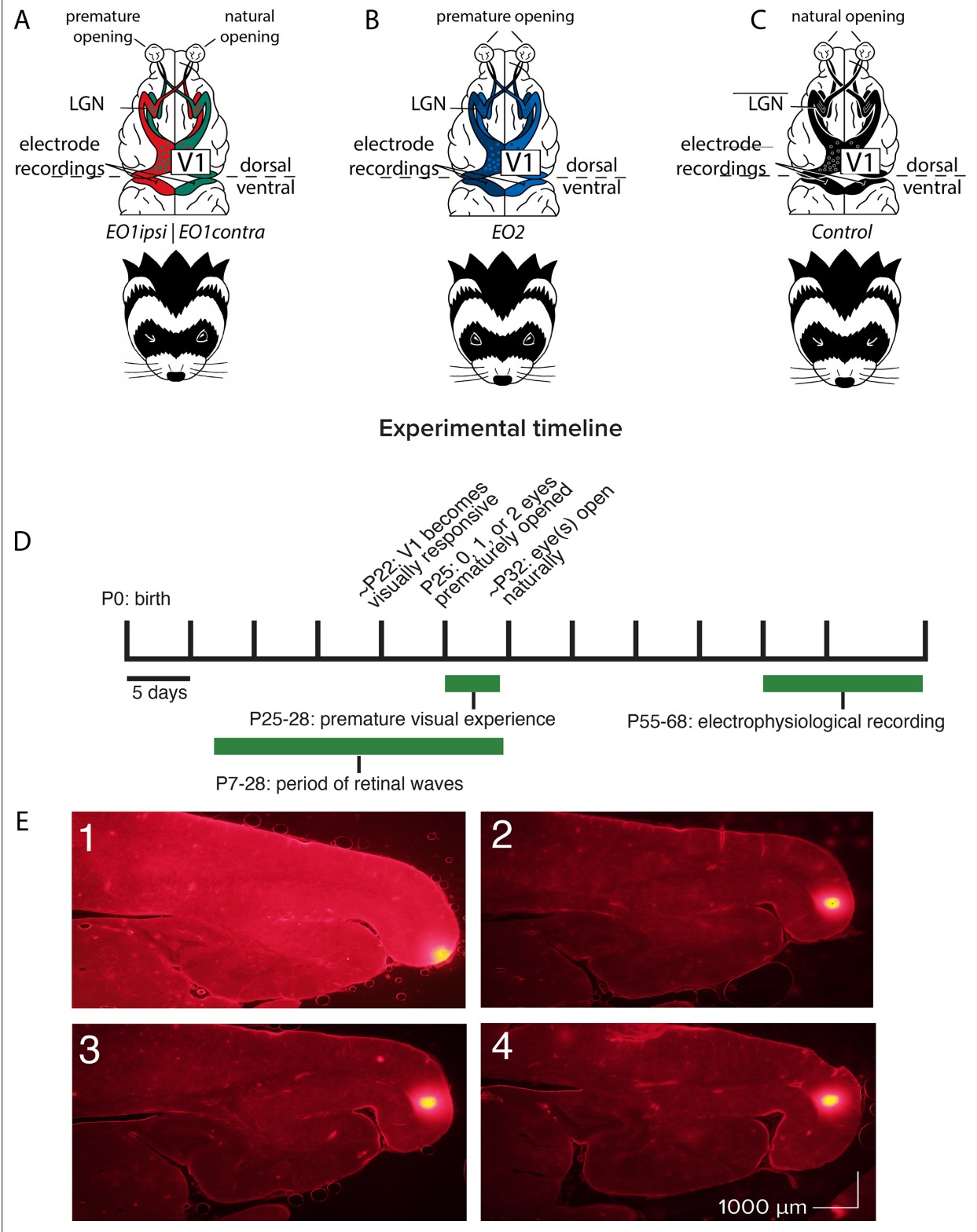

**Figure 1.** Recording schematics, experimental timeline, and electrode track reconstructions. (**A**) Recording schematics: EO1contra cells were those recorded in monocular cortex contralateral to a single prematurely opened eye. EO1ipsi cells were those recorded in monocular cortex ipsilateral to a single prematurely opened eye. Dots schematize ocular dominance columns. (**B**) EO2 cells were those recorded in monocular cortex after both eyes were opened prematurely. (**C**) Control cells were those recorded in monocular cortex after both eyes were allowed to open naturally. (**D**) Experimental timeline. Animals that experienced premature vision had one eye or both eyes opened on P25. All animals experienced a period outside of the nest

*Figure 1 continued on next page*

*Figure 1 continued*

in the form of daily 2 hr handling sessions beginning on P25 and ending on P28, allowing unguided viewing of the laboratory environment through prematurely opened eyes or low resolution experience through closed lids. Electrophysiological recording took place between P55-68, at a time when the critical period for ocular dominance was closing or closed (*Issa et al., 1999*). (**E**) Electrode track reconstructions showing penetrations (bright spots at right/posterior) along the posterior cortical surface, in monocular V1 (*Law et al., 1988*; *White et al., 1999*). Following the recording, the recording electrode was removed and replaced with a marking electrode coated in DiI.

## Orientation and direction tuning index values are slightly increased by premature experience

Cells in all animals exhibited some degree of orientation and direction selectivity. Orientation selectivity was assessed for stimuli that moved back and forth in two opposite directions at each cell's preferred spatial frequency. Direction selectivity was assessed at the neuron's optimal temporal frequency and a spatial frequency of 0.1. Example responses for each group are shown in *Figure 2A–H*. Control animals exhibited a baseline average orientation index value (1-CV) of 0.42, and this value was not significantly elevated in EO1ipsi neurons (0.45, p<0.54, Linear Mixed Effects Model). However, orientation selectivity was slightly elevated in EO1contra neurons (0.52, p<0.024) and EO2 neurons (0.54, p<0.016, LMEM). Direction selectivity as assessed by 1-DCV was slightly reduced in EO1ipsi neurons (0.24, p<0.027, LMEM) as compared to control values (0.29), and 1-DCV values were not significantly different in EO1contra neurons (0.29, p<0.296) or EO2 neurons (0.33, p<0.169, LMEM). The overall distribution of direction preferences in 45° bins did not significantly differ across the groups (chi-square test, p=0.11), nor did the distribution of orientation angle preferences in 45° bins (chi-square test, 5.3x10–2). Therefore, we observed only minor long-term changes in orientation and direction tuning between animals with premature vision and control animals.

## Temporal frequency tuning is aberrant in animals with premature visual experience

We were struck by the unusual temporal frequency tuning curves that we observed in animals whose eyes were opened prematurely. Many EO1contra cells exhibited strong responses at the lowest temporal frequency tested (*Figure 3A*), and we observed a trend for this in EO2 cells as well (*Figure 3B*). Further, several cells also showed evidence of a suppression of the response below baseline. Tuning curves from EO1ipsi cells (*Figure 3C*) superficially looked more like curves from control animals (*Figure 3D*), although quantitative analyses uncovered small differences. Most cell types exhibited similar peak temporal frequency values (*Figure 3E*), except that EO1ipsi cells exhibited slightly higher temporal frequency preferences (p<0.021, LMEM). To quantify the response to the lowest temporal frequency tested, we calculated a low-pass index, defined as the ratio of the response to the lowest temporal frequency tested to the largest temporal frequency response observed (*Figure 3F*). Temporal frequency low-pass index values of control cells were 0.20 on average, while EO1contra cells showed low-pass index values that averaged 0.35 (p<0.039, LMEM) and EO2 cells showed empirically elevated values of 0.33 that were not significantly different from control (p<0.142, LMEM). High pass index values, determined by calculating the response at the highest temporal frequency tested divided by the maximum response, did not differ across these animals (*Figure 3G*).

In order to assess tuning bandwidth, we calculated low- and high-frequency cutoff values by identifying the low- and high-frequency values where the response dropped below half the maximum response value. If this never occurred, the cell was considered to be low-pass (if this never occurred on the low-frequency side) or high-pass (if this never occurred on the high-frequency side). We then calculated the bandwidth between this low and high value in octaves. If a cell was low-pass or high-pass or both, the cell was said to have infinite bandwidth. In *Figure 3H*, we show bandwidth computed from rectified responses and saw empirically elevated bandwidths in EO1contra cells (2.33 octaves, p<0.07) and EO2 cells (2.33 octaves, p<0.08) compared to control cells (1.95 octaves) or EO1ipsi cells (1.91 octaves, p<0.937). Many cells exhibited suppression, where the response fell below the response to a control stimulus, and we took the absolute value of the response in calculating the absolute bandwidth so that both positive and negative responses were considered. The median absolute bandwidth (*Figure 3I*) of control cells (4.1 octaves) was narrower than any of the cells in animals whose eyes were opened early, including EO1ipsi cells (4.7 octaves, p<0.0013), EO1contra cells (4.8 octaves, p<0.002),

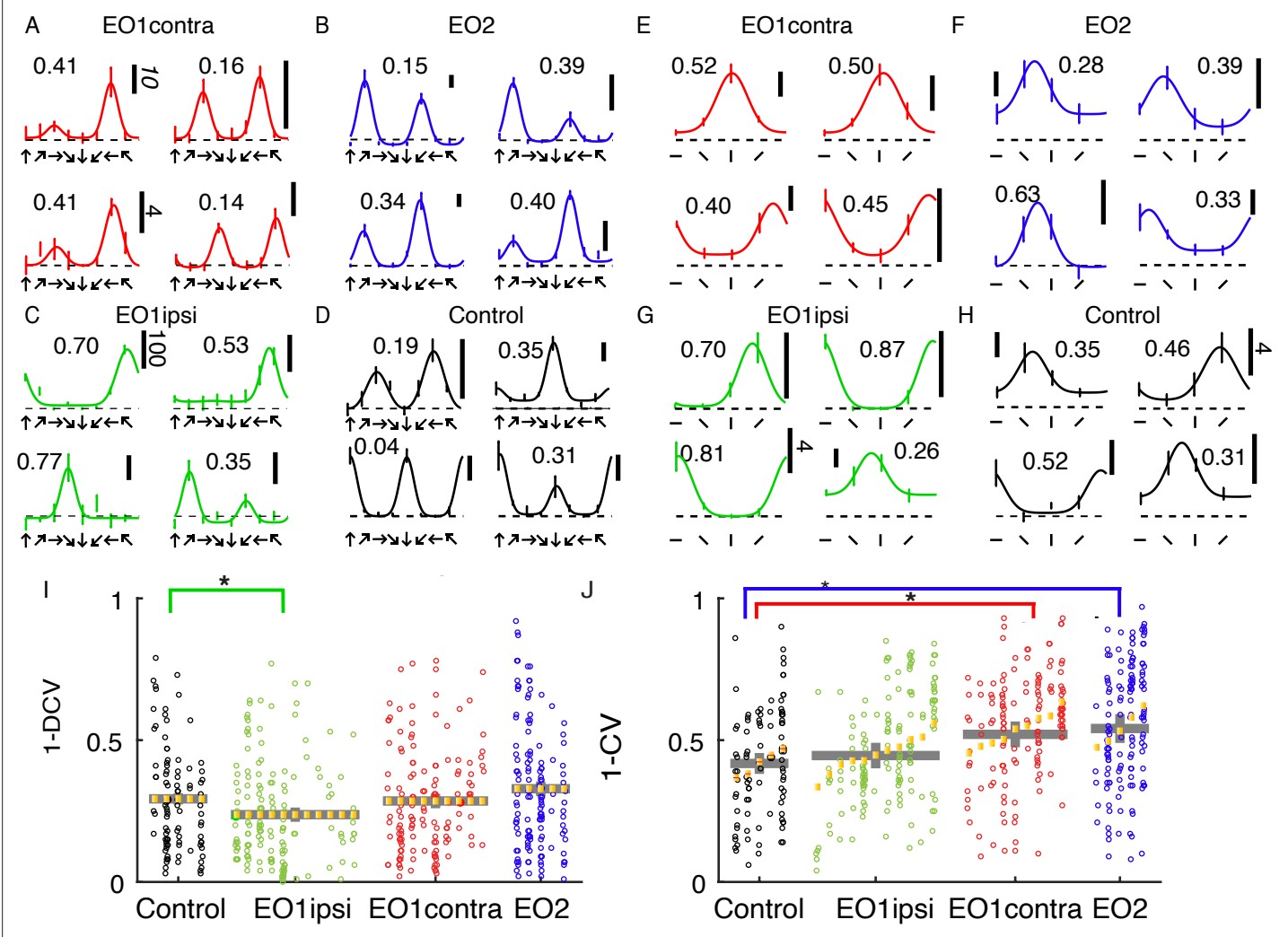

**Figure 2.** Early eye opening produced only minor changes in orientation and direction selectivity index values: (**A**) Example direction tuning curves of individual EO1contra cells measured after the close of the critical period. Direction tuning curves are shown for stimulation at the optimal temporal frequency. Direction tuning curves look coarsely normal. Vertical scale bar represents 10 spikes per second (standard indicated in italics) unless otherwise indicated. The dashed line shows mean response to blank / control stimuli. Number indicates 1-DCV value. (**B**) Same, for EO2 cells (**C**) Same, for EO1ipsi cells: (**D**) Same, for control cells (**E**) Orientation tuning curves of individual EO1contra cells, measured in different epochs and in some different animals than the direction tuning curves. Temporal frequency was 2 Hz, and the spatial frequency was optimal for each cell. Numbers indicate orientation selectivity index values 1-CV. (**F**) EO2 cells G: EO1ipsi cells (**H**) control cells (**I**) Results of a linear mixed effects analysis of direction selectivity index values, as assessed by (1-DCV), for all cells and conditions in the study. Each small column shows the cells for an individual animal. Yellow lines indicate random effect values for each animal. Gray lines indicate condition means and standard errors of the mean from the linear mixed effect model. Animals within a group have been sorted by mean index value. Condition coefficients that differ significantly from 0 are indicated with * (p<0.05), and conditions compared are indicated by comparison bars. See text for p values. (**J**) Same for orientation selectivity index values (1-CV) . EO1contra and EO2 cells exhibited slightly higher orientation selectivity index values than control animals, while EO1ipsi cells exhibited slightly lower direction selectivity values than other conditions.

and EO2 cells (5.1 octaves, p<0.00015, LMEM). These changes in bandwidth were related to changes in low- and high-cutoff values (*Figure 3—figure supplement 1*).

## Spatial frequency tuning is relatively unaffected by premature visual experience

Spatial frequency tuning was hardly influenced by early eye opening. We assessed spatial frequency at a cell's preferred orientation, as determined by responses to drifting gratings that moved back and forth in opposite directions. Example tuning curves are shown in *Figure 4A–D*.

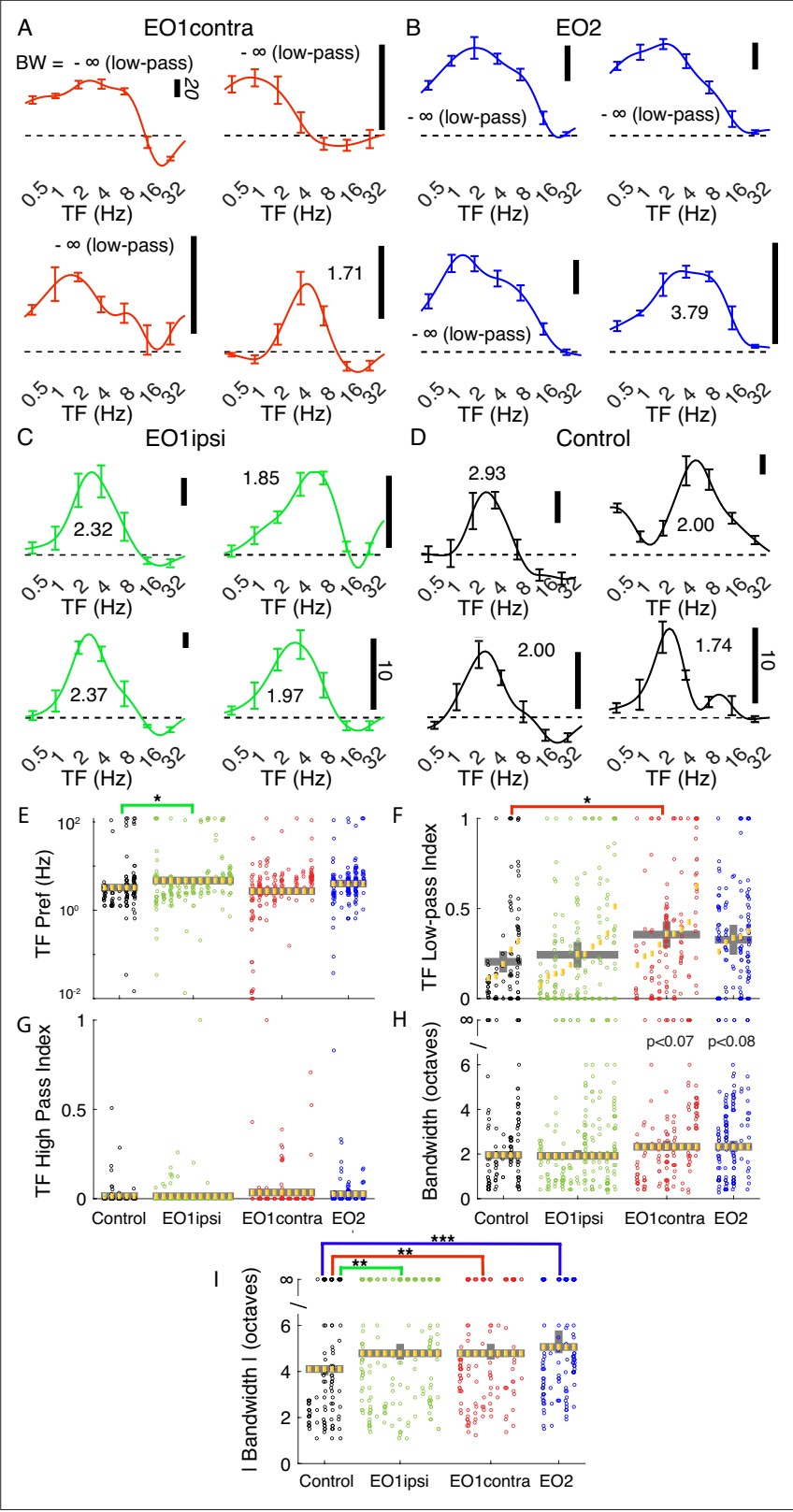

**Figure 3.** Premature eye opening altered temporal frequency tuning after maturity. Representative temporal frequency tuning curves recorded from single cells: (**A**) EO1contra: cells contralateral to a single prematurely opened eye. Several cells exhibited strong responses at the lowest temporal frequency tested (that is, several exhibited low-pass tuning). Temporal frequency was reported at each cell's overall preferred direction (assessed

*Figure 3 continued on next page*

*Figure 3 continued*

over all temporal frequencies). Number indicates bandwidth in octaves. -∞ is low-pass. Vertical scale bar indicates firing rate (20 spikes/sec standard in italics unless otherwise noted). (**B**) EO2: cells recorded in an animal with both eyes prematurely opened. Again, many cells exhibited low-pass responses. (**C**) EO1ipsi: cells ipsilateral to a single prematurely opened eye. (**D**) Control: cells recorded in animals that opened both eyes naturally. (**E**) Linear mixed effects model plot of median temporal frequency preference; linear mixed effects plotted as in *Figure 2*. Median temporal frequency preference was slightly higher in EO1ipsi cells than other cells. (**F**) Temporal frequency low-pass index values were significantly elevated in EO1contra cells, indicating stronger responses to the lowest temporal frequency tested. (**G**) No differences were observed in high pass index values. (**H**) Bandwidth of the rectified response. (**I**) Bandwidth of the absolute value of responses. Cells in animals that experienced early eye opening exhibited wider bandwidths (less selectivity) than control animals. See text for numbers and p-values.

The online version of this article includes the following figure supplement(s) for figure 3:

**Figure supplement 1.** Premature eye opening alters L50 and H50 values.

In linear mixed effects modeling, no significant differences were observed in peak spatial frequency preferences across these groups (control: 0.031 cpd; EO1ipsi: 0.045 cpd, p<0.19; EO1contra: 0.031 cpd, p<0.949; EO2: 0.042 cpd, p<0.362, LMEM). Similarly, median spatial frequency low-pass index values were very similar, with a slightly significant change noted for EO1ipsi animals (control: 0.86; EO1ipsi: 0.76, p<0.0499; EO1contra: 0.85, p<0.928; EO2: 0.79, p<0.390, LMEM). We observed slightly higher spatial frequency high pass index values in EO2 cells, but effects were modest: (control: 0.02; EO1ipsi: 0.02, p<0.857; EO1contra: 0.04, p<0.090; EO2: 0.06, p<0.0323, LMEM).

We observed no significant effects with respect to spatial frequency bandwidth, either when measured with responses that were greater than 0 or measuring the absolute response (that is, interpreting responses below baseline as positive responses). Bandwidths of individual cells were often infinite, given the large preponderance of low-pass cells, and so we analyzed the data in rank order instead of the raw values. Median values for both positive-only responses and absolute responses were always above the ranks of cells with finite bandwidths, indicating that the typical cell in all groups had infinite bandwidth. Differences from control were not significant for positive-only responses (EO1ipsi: p<0.310, EO1contra: 0.622, EO2: p<0.898, LMEM) or absolute responses (EO1ipsi: p<0.06, EO1contra: p<0.597, EO2: p<0.385, LMEM).

In short, we found very little evidence of substantial changes to spatial frequency tuning in these monocularly-driven cells.

## Spontaneous firing rates and response suppression are increased in animals that have had premature visual experience

Hebbian and homeostatic mechanisms are active early in development (*Turrigiano and Nelson, 2004*) and may be critical for setting response gains and the background excitability of neural circuits (*Roy et al., 2018*; *Roy et al., 2020a*). To examine whether response gains or excitability were impacted by premature vision, we quantified maximum firing rates in response to visual stimuli, response suppression, and background firing rates.

The peak mean responses of neurons to visual stimuli (*Figure 5A*) did not differ among the experimental groups. Control neurons exhibited average peak responses of 7.0 Hz; EO1ipsi neurons did not differ significantly (6.4 Hz, p<0.600), nor did EO1contra neurons (10.1 Hz, p<0.518) or EO2 neurons (10.5 Hz, p<0.295).

In contrast to peak responses, response suppression below background levels was pronounced in animals with prematurely opened eyes (*Figure 5B*). We calculated all responses by subtracting the background rate measured during control stimuli (gray screen) and took response suppression for each cell to be the maximum amount of suppression below background rates for the average response to any visual stimulus. Suppression was rarely observed in control cells (average: 2.1 Hz), but was substantially elevated in EO1ipsi cells (7.0 Hz, p<1.59e-9) and slightly elevated in EO1contra cells (3.4 Hz, p<0.022) and EO2 cells (2.8 Hz, p<0.035).

Finally, background firing rates were significantly higher in animals whose eyes were opened early (*Figure 5C*). Spontaneous firing rates were low in control animals (2.6 Hz) and were markedly increased in EO1ipsi hemispheres (11.1 Hz, p<1.09e-4), EO1contra hemispheres (6.6 Hz, p<0.0132),

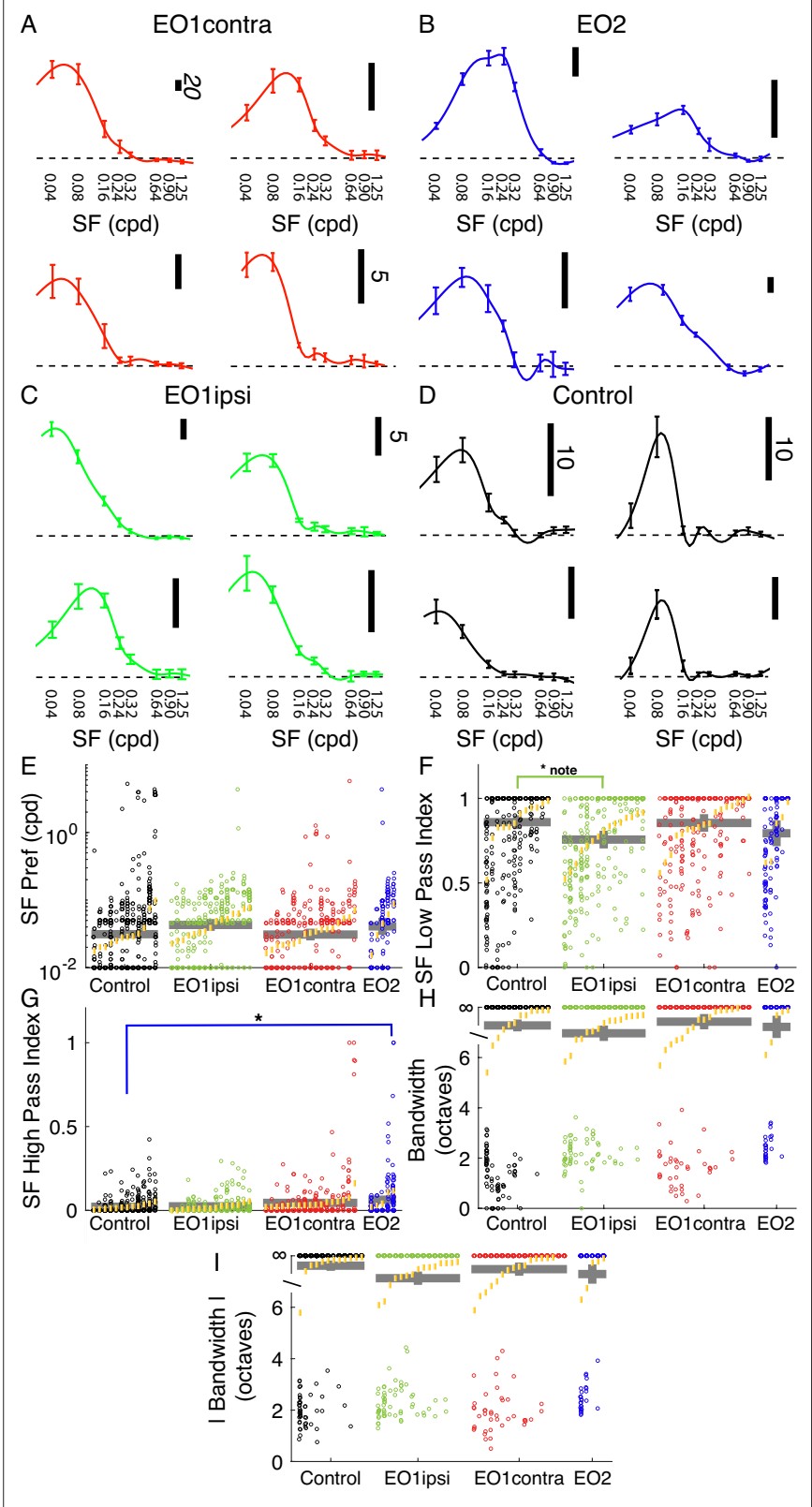

**Figure 4.** Premature eye opening had very small impacts on spatial frequency tuning. (**A**) Example spatial frequency tuning curves in EO1contra cells. (**B**) Same for EO2 cells. (**C**) Same for EO1ipsi cells. (**D**) Same for control cells. (**E**) Spatial frequency peak preferences for all cells, animals, and experimental groups (same linear mixed effect model plot layout as *Figure 2*). (**F**) Spatial frequency low-pass index values. EO1ipsi cells exhibit a

*Figure 4 continued on next page*

*Figure 4 continued*

slightly significant (p<0.499) decrease in low-pass index but differences across groups were small. (**G**) High pass index values for spatial frequency; differences were modest across groups although cells with strong high-pass responses were more common in EO2 cells. (**H**) Spatial frequency bandwidth, in octaves. Most cells exhibited low-pass profiles and had infinite bandwidth, and no differences were noted across the groups. (**I**) Spatial frequency bandwidth for absolute response.

and EO2 hemispheres (6.6 Hz, p<0.0248). Therefore, early opening produced a long-lasting effect on spontaneous firing rates after the critical period.

In all, early eye opening induced changes in response suppression and background firing rates that were still evident at the time of our measurements several weeks later. We observed these changes in both hemispheres, which indicates that the differences in direct visual drive to visual receptive fields are not underlying these differences but instead are the result of some more global process. These results are consistent with the idea that early eye opening induces long-term changes to circuit-level excitability that are not corrected during the remainder of development.

## Discussion

We explored the impact of premature vision on the development of a number of V1 RF properties in ferrets. We found small increases in direction and orientation tuning in cells that viewed the world through prematurely opened eyes. We observed profound changes in temporal frequency tuning: cells recorded from animals that had their eyes opened prematurely exhibited more low-pass temporal frequency tuning curves and broader temporal frequency bandwidths compared to controls. Spatial frequency preferences were not substantially altered. Finally, cells in either hemisphere in animals with prematurely opened eyes had higher spontaneous firing rates and exhibited marked suppression below this spontaneous firing rate in response to some visual stimuli.

### A novel temporal frequency tuning deficit with possible implications for motion processing

We found that ferrets with premature visual experience exhibited a pronounced broadening of temporal frequency tuning curves and increases in the fraction of low-pass neurons. The alterations in temporal frequency tuning identified in ferrets following premature eye opening are strongly suggestive of a deficit in motion processing. The broadening of temporal frequency tuning curves and increase in the proportion of low-pass cells in V1 could translate to a decrease in the animal's ability to distinguish the temporal frequency (and perhaps the speed) of a moving visual stimulus.

Altered rearing has been shown to influence the subsequent development of motion processing in various ways. Dark-reared cats and ferrets exhibit poor direction selectivity (*Imbert and Buisseret, 1975*; *Li et al., 2006*), but dark-reared ferrets do not show strong deficits in temporal frequency processing. Overall responses after dark-rearing are reduced, but temporal frequency tuning is only slightly narrowed (*Li et al., 2006*). Cats raised in a strobe-light environment (mostly after eye opening) exhibited strong changes in subsequent direction selectivity (*Kennedy and Orban, 1983*; *Humphrey and Saul, 1998*) and behavioral sensitivity to motion (*Pasternak et al., 1981*; *Pasternak et al., 1985*) that partially recovers with motion detection training. However, temporal frequency tuning of these animals has not been reported in detail. *Pasternak et al., 1981* reported that strobe-reared cats exhibited greater difficulty in distinguishing slow-moving stimuli from static stimuli compared to controls, an ability that slightly improved with practice, suggesting possible temporal frequency deficits. *Kennedy and Orban, 1983* report a reduction in high-pass velocity-tuned cells, but not an increase in low-pass cells; *Humphrey and Saul, 1998* do not report temporal frequency tuning across their population, but their example cell does appear to be broadly tuned and low-pass. Temporal frequency tuning is not subject to rapid modification shortly after eye opening, as Ritter and colleagues demonstrated that providing ferrets that had just opened their eyes naturally with 6–9 hr of visual stimulation of a single temporal frequency did not elicit a shift in temporal frequency tuning (*Ritter et al., 2017*). The same stimulation causes a dramatic increase in direction selectivity (*Ritter et al., 2017*) that normally develops in the first week to 10 days after eye opening and requires visual experience (*Li et al., 2006*; *Li et al., 2008*; *Van Hooser et al., 2012*; *Roy et al., 2020b*). The only manipulation we know of in the

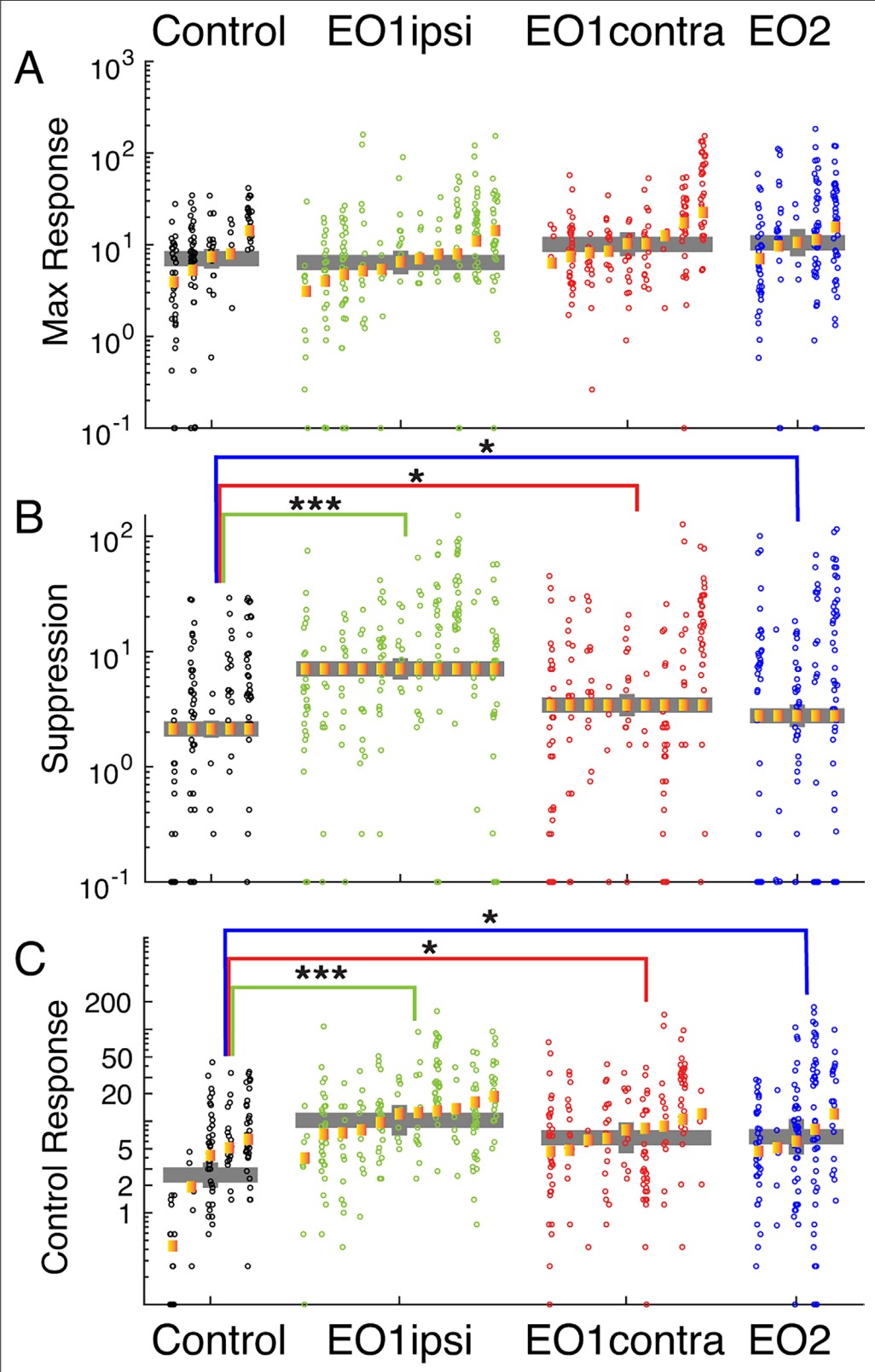

**Figure 5.** Premature eye opening altered response suppression below baseline and spontaneous activity of V1 neurons. (**A**) Maximum firing rates of cells to visual stimulation were unchanged across experimental conditions. (**B**) Response suppression rates, defined as the largest reduction in responses to visual stimulation below background firing rates, were substantially bigger in EO1ipsi, EO1contra, and EO2 cells, indicating that responses

*Figure 5 continued on next page*

*Figure 5 continued*

below baseline were features of the stimulus response in many of these cells. (**C**) Background firing rates. Responses of cells during presentation of control stimuli, where the screen remained gray for the same duration as our typical visual stimuli. EO1ipsi, EO1contra, and EO2 cells showed greatly and significantly elevated firing rates compared to controls. See text for numbers.

---

literature that has been shown to modify temporal frequency properties of developing animals is prey capture: mice that learned to hunt crickets in the critical period for ocular dominance plasticity exhibit lower low-pass index values (*Bissen et al., 2025*).

In sum, the deficits in temporal frequency tuning we observed are unlike those reported previously. Premature vision produces permanent alteration of temporal frequency tuning curves, while dark-rearing does not, and comparable results from strobe rearing are not available.

## Biology of early development in mustelids

In the wild, mustelids raise their young in nests in the ground, in cavities such as holes in trees or caves, or in areas of dense vegetation (*Ruggiero et al., 1994*). They may move the young from one nest to another as they grow, but otherwise, the young are primarily in the relatively dark nest. It is highly likely that some light penetrates and that information about the 24 hr cycle is available, but the light is likely to be dim and unlikely to provide a basis for high luminance, high contrast stimulation through the closed lids. The animals begin to spend substantial time outside the nest after eye opening.

The ferret is a domesticated strain of the European polecat. In laboratory settings, ferret jills give birth and keep their kits in a nest box. A laboratory typically maintains a 24 hr cycle with 12 or 14 hr of light, and the light reaching the closed lids must first pass through the cage, the nest box, and the nesting material. Therefore, developing ferrets have an obvious circadian light signal, but the light available for image formation is likely dim and of low contrast.

Although the light that reaches the close lids in developing ferrets is likely to be relatively dim, and any image-forming signal passing through the closed lids would be highly filtered in luminance, spatial frequency, and contrast, it is important to remember that visual input before natural eye opening (through the closed lids) can drive activity in retina, LGN, and cortex (*Huttenlocher, 1967*; *Chapman and Stryker, 1993*; *Krug et al., 2001*; *Akerman et al., 2002*; *Akerman et al., 2004*). Further, orientation selectivity can be observed through the closed lids (*Krug et al., 2001*), indicating that some coarse image-forming information does make it through the closed lids.

## Mechanisms underlying altered receptive fields

One way that premature patterned vision could alter the development of the brain is by altering the character of retinal waves. The low-resolution visual input before natural eye opening is capable of driving activity in the visual circuit (*Huttenlocher, 1967*; *Chapman and Stryker, 1993*; *Krug et al., 2001*; *Akerman et al., 2002*; *Akerman et al., 2004*), and evidence shows that this early form of low-resolution visual experience plays a role in the development of ON-OFF segregation in dLGN (*Akerman et al., 2002*) and the initiation of retinal waves (*Tiriac et al., 2018*). Propagation of high luminance, high contrast, and high spatial frequency input through the developing retina may alter or disrupt the typical retinal wave formation and propagation, and thus may have deleterious effects on developmental processes driven by endogenous activity. Alternatively, high-contrast signals may not interrupt retinal waves but may contribute independent activity that, through experience-dependent plasticity mechanisms, produces the alterations in receptive field properties that we observed.

Several other basic mechanistic questions remain unanswered. It is unclear where in the visual circuit cascade these deficits first arise. Does the lateral geniculate nucleus or retina exhibit altered temporal frequency tuning? Is the influence of the patterned visual stimulation instructive, so that if one provided premature stimulation with only certain temporal frequencies, one would see selectivity for those temporal frequencies, or would tuning always be broad? Other questions remain concerning the top-down influence on V1 from 'higher' motion areas such as MT (monkeys) or PSS (ferret); MT exhibits mature neural markers earlier than V1 (*Bourne and Rosa, 2006*), and suppression of PSS impacts motion selectivity in V1 (*Lempel and Nielsen, 2021*). Finally, our recordings were performed in monocular cortex so that we could be sure of the developmental condition of the eye that drove the classic responses. It is interesting to speculate about what might occur more centrally in binocular

visual cortex. Ocular dominance shifts are not induced when one eye is opened prematurely (*Issa et al., 1999*), indicating that ocular dominance plasticity is not engaged at this early stage, but one might imagine that the impacts on temporal frequency and spontaneous firing rates would still be present. Future studies will be needed to address these questions.

We introduced premature patterned vision at a time when cortical inhibition is undergoing substantial changes. GABAergic signaling has already undergone its switch (*Ben-Ari, 2002*) from providing primarily depolarizing input to hyperpolarizing input by P21-23 (*Mulholland et al., 2021*). In the days prior to eye opening, inhibitory cells exhibit activity that is closely associated with the emerging functional modules that will reflect orientation columns (*Mulholland et al., 2021*), but do not yet exhibit selectivity to orientation, in contrast to excitatory neurons, which do exhibit selectivity to orientation at that time (*Chang and Fitzpatrick, 2022*).

The broad tuning in temporal frequency and elevated spontaneous firing rates that we observed could reflect a deficit in cortical inhibition in animals that experienced premature vision. Yet selectivity was not entirely destroyed, as neurons still exhibited selectivity for orientation and direction. Selectivity can be achieved with a combination of amplification of appropriate excitatory signals and inhibitory suppression that is either selective or broad (*Ben et al., 1995*; *Rubin et al., 2015*; *Somers et al., 1995*; *Hatta et al., 1998*), so it is unclear how the circuit has changed. The amount and timing of inhibition, whether feed-forward or feedback, could be altered.

## Alterations in ipsilateral receptive fields

One of our most surprising findings was a difference in the receptive fields and firing properties of cells that were ipsilateral to the early opened eye. It is straightforward to imagine how Hebbian mechanisms might influence monocular cells that are contralateral to the early-opened eye, but one would have to imagine different and possibly more global mechanisms to explain changes in monocular cells that view the world through the eye that opened naturally.

Homeostatic mechanisms such as synaptic scaling (*Desai et al., 1999*, *Turrigiano et al., 1998*), and regulation of intrinsic excitability (*Desai et al., 1999*; *Hengen et al., 2013*) operate during the critical period for ocular dominance plasticity and serve to keep ongoing firing rates within a cell-defined range (*Hengen et al., 2013*; *Hengen et al., 2016*). It is possible that these homeostatic mechanisms are not yet operating in the presensory period, and that the extra activity causes an increase in excitatory forces (whether synaptic or intrinsic) relative to inhibitory forces.

Alternatively, opening a single eye early may still influence feed-forward activity in the ipsilateral hemisphere. Work in ferrets and rats suggests a role for retinal waves in the development of inter-hemispheric connectivity; bilateral enucleation of ferrets induces anomalies in the distribution of callosal cells in primary visual cortex (*Bock et al., 2012*), and results in callosal projections connecting topographically mismatched loci in VC in P4-6 rats (*Olavarria and Hiroi, 2003*). Further, retinal waves demonstrate coordinated activity between both hemispheres in mice and ferrets (*Weliky and Katz, 1999*; *Ackman et al., 2012*; *Ackman and Crair, 2014*). Recent findings describe a transient retina to retina connection in mice and ferret, which exists during the period of retinal waves (*Murcia-Belmonte et al., 2019*). This connection provides a plausible mechanism for the theorized inter-retinal coordination of waves thought to be necessary to drive development of bilateral topographic maps (*Adams and Horton, 2003*; *Murcia-Belmonte et al., 2019*). Given that light is capable of initiating retinal waves (*Tiriac et al., 2018*), it is feasible that exposure to light through a single, prematurely opened eye would initiate retinal waves capable of propagating to the closed eye. Given the qualitative differences between vision through a closed lid vs open eye, initiation and propagation of retinal waves would be both asymmetric and mutually influential between retinas, potentially accounting for the differences in RFs observed between EO1ipsi and control hemispheres.

## Comparison with prior studies

There have been only a few animal studies that have sought to introduce inappropriately early vision. Lichliter and colleagues (*Lichliter, 2018*; *Lichliter, 2000*; *Sleigh and Casey, 2014*) prematurely opened the shells of bobwhite quail eggs and provided visual experience to the chick and found a subsequent loss of visual selectivity to bobwhite hens vs. hens of another species (*Sleigh and Lichliter, 1995*). Further, they demonstrated multi-modal impacts on auditory processing after premature visual experience, in that animals failed to eventually become responsive to cross-species hen calls. In behavioral

studies of rats whose eyes were prematurely opened 7 days early, investigators found that opening a single eye (but not opening both eyes early) caused a deficit in pivoting locomotor reactions on the side of the early-opened eye; that is, rats pivoted more frequently towards stimuli viewed through the eye that opened on time (*Foreman and Altaha, 1991*). To our knowledge, the present work is the first to study cortical receptive fields after premature sensory experience. All of these studies, including our own, have noted deficits after premature experience.

## Premature vision as a risk factor for visual dysfunction

Very premature human babies exhibit a variety of visual deficits later in life that are not explained by the condition of the eye, broadly termed Cortical/Cerebral Visual Impairment (CVI; *Ortibus et al., 2009*; *Kozeis, 2010*; *Dutton, 2013*; *Sakki et al., 2018*; *Pamir et al., 2021*). These very premature babies are a heterogeneous population. Some of these infants have suffered clear brain injuries (*Slidsborg et al., 2012*; *Dutton, 2013*) of various types, such as hypoxia, brain bleeding, or periventricular leukomalacia (*Gallo and Lennerstrand, 1991*; *Hokken et al., 2023*, *Kozeis, 2010*), and these injuries clearly have a direct impact on subsequent visual deficits that go beyond any influence of early experience. But studies that have excluded individuals with known brain damage or retinal problems (or analyzed these populations separately) have also found that extreme prematurity itself has an impact on later vision, particularly on motion perception (*Atkinson and Braddick, 2007*; *Benassi et al., 2018*; *Guzzetta et al., 2009*; *Hou et al., 2011*; *MacKay et al., 2005*) and sometimes lowered visual acuity (*Jain et al., 2022*). Previous investigators have speculated that inappropriately early visual stimulation may contribute to these deficits, as in *Taylor et al., 2009*: "Why the dorsal stream should be particularly vulnerable in this population is not completely understood. One possibility is that the unusually early visual stimulation that very premature infants are exposed to may, in itself, have a differential effect on the functional development of the dorsal and ventral streams". Our data are consistent with the idea that inappropriately early visual stimulation contributes to deficits in motion processing. A moving object has both a speed and direction, and temporal frequency is critical to determine an object's speed. The major temporal frequency tuning deficits we have observed in receptive fields are consistent with a motion deficit.

## Premature vision as a cause of visual plasticity

Regardless of whether premature patterned vision has a clinically relevant role for human vision, the experiments here demonstrate clearly that premature patterned vision has a long-lasting impact on the development of receptive fields in the visual cortex. The results suggest that the evolved late eye opening of many rodents and carnivores, and the prolonged prenatal period of primates, may help to protect the visual system from exposure to high luminance, high contrast, and high spatial frequency stimulation until the visual system is ready for this type of stimulation. This shielding of the developing visual system is not found in all vertebrates, as tadpoles and zebrafish have transparent eyelids that permit patterned vision from very early stages (*Avitan et al., 2017*; *Demas et al., 2012*). By altering premature patterned vision, it may be possible to uncover the plasticity rules that are at play in the developing mammalian visual system before the natural onset of vision.

# Materials and methods

**Key resources table**

| Reagent type (species) or resource | Designation | Source or reference | Identifiers | Additional information |
|---|---|---|---|---|
| Biological sample (Ferret) | Ferret | Marshall Bio-Resources | Mustelo putorius furo | Female ferrets used |
| Antibody | Rabbit anti-NeuN, Alexa Fluor 488 conjugated | Millipore | Cat# ABN78A4 | Dilution: 1:300 |
| Chemical compound, drug | Ketamine | Patterson Veterinary | 07-890-8598 | 20 mg kg$^{-1}$ im |
| Chemical compound, drug | Isoflurane Covetrus | Covetrus | 029405 | 1.5–3% in $N_2O/O_2$ mixture for surgery |
| Chemical compound, drug | Isoflurane Covetrus | Covetrus | 061843 | Infused into wound margins |

*Continued on next page*

*Continued*

| Reagent type (species) or resource | Designation | Source or reference | Identifiers | Additional information |
|---|---|---|---|---|
| Chemical compound, drug | Dexamethasone | Patterson Veterinary | 07-808-8194 | 0.5 mg kg⁻¹ im |
| Chemical compound, drug | Atropine | Patterson Veterinary | 07-869-6061 | 0.16–0.8 mg kg⁻¹ im |
| Chemical compound, drug | Gallamine triethiodide | Sigma Aldrich | G8134-25G | 10–30 mg kg⁻¹ h⁻¹ |
| Chemical compound, drug | Sodium pentobarbital (Euthasol) | Patterson Veterinary | 07-805-9296 | 200 mg/kg, IP |
| Chemical compound, drug | DiI | Sigma Aldrich | 42364–100 MG | Used for electrode track reconstruction (*DiCarlo et al., 1996*) |
| Chemical compound, drug | Paraformaldehyde | Sigma Aldrich | P6148-1KG | 4% in 0.1 M PBS |
| Chemical compound, drug | Triton-X 100 | Sigma Aldrich | 9002-93-1 | 0.3% in PBS |
| Chemical compound, drug | Fluoromount-G | Southern Biotech | 0100–20 | |
| Software, algorithm | MATLAB | MathWorks | RRID:SCR_001622 | Used for stimulus creation and data analysis |
| Software, algorithm | Psychophysics Toolbox | *Brainard, 1997*; *Pelli, 1997* | RRID:SCR_002881 | Used for visual stimuli display |
| Software, algorithm | Spike2 | Cambridge Electronic Design | RRID:SCR_000903 | Used for stimulus timing acquisition |
| Software, algorithm | JRClust | *Jun et al., 2017* | | Used for offline spike sorting in Matlab |
| Software, algorithm | fitlme | Matlab | N/A | Used for linear mixed-effects modeling |
| Software, algorithm | Neuroscience Data Interface (NDI) | *García Murillo et al., 2022* | RRID:SCR_023368 | Data management and sharing |
| Other | Multichannel electrodes | Plexon | Plexon S probes | 32 channels, 50 µm spacing |
| Other | Amplifier/Digitizer | Intan Technologies | RHD2000 system | |
| Other | Data acquisition board | Cambridge Electronic Design | Micro1401 | |
| Other | Manipulator | Sutter Instruments | MP-285 | |
| Other | CRT Monitor | Sony | GDM-520 | 21-inch, 800x600, 100 Hz |
| Other | Ophthalmoscope | Heine | Heine Omega 600 | |
| Other | Ophthalmic Lens | Volk | 78D or 90D lens | |
| Other | Sliding Microtome | Leica | SM2010R | |
| Other | Fluorescent Microscope | Keyence | BX-Z 710 | |
| Other | Brandeis Light Microscopy Core Facility | Brandeis University | RRID:SCR_025892 | Houses SM2010R and BX-Z 710 |

All experimental procedures were approved by the Brandeis University's Institutional Animal Care and Use Committee (IACUC) and performed in compliance with National Institutes of Health guidelines.

## Animal source and housing

Ferrets (*Mustelo putorius furo*) were obtained from Marshall Bio-Resources. Litters of 4 or more kits arrived with a jill between postnatal days (P) 12–21. Animals were housed in a room with timed lights (12 hr on, 12 hr off) in a custom stainless-steel cage (60 cm × 60 cm×35 cm) with a hammock and small toys. For the entire study, a total of 27 female ferrets were used and all experimental procedures were carried out between postnatal days P55–68. Female ferrets were used because housing mature male ferrets in the same room with mature female ferrets causes stress to the female ferrets.

## Facilitating premature visual experience

We divided animals into three experimental groups. The first group of animals had one of their eyes gently opened with forceps at P25, while the other eye was left closed to open naturally, and time of

natural eye opening was noted. We used the label early Eye Opening One Contralateral (henceforth EO1contra) to indicate neurons in the monocular visual cortex that viewed the world through the (contralateral) eye that was opened early, and early Eye Opening 1 Ipsilateral (henceforth EO1ipsi) to indicate neurons in the monocular visual cortex that viewed the world through the eye that opened on time. In the second group of animals, both eyes were opened at P25, and neurons from the monocular visual cortex that viewed the world through either of these eyes were labeled early Eye Opening 2 (henceforth EO2). Finally, the third group of animals was two eye controls and was allowed to open both eyes normally, with the time of natural eye opening noted. Ferrets frequently open one eye up to a day before the other, and any disparity in the time of opening between the eyes was noted. Ferret kits in laboratory housing receive limited visual stimulation through their closed lids, as the mother actively keeps the kits in their relatively dark nest. In order to ensure that animals with early-opened eyes actually had patterned visual experience (and animals with closed lids had the same stimulation filtered through the lids), animals were brought to the lab for 2 hr a day for 4 consecutive days beginning at P25. These ferret kits were placed in a rat cage atop a heating pad and gently manipulated to maintain wakefulness during this natural, unguided viewing. We studied 17 animals that had one eye prematurely opened, 4 animals that had both eyes opened, and 11 controls.

## Surgical preparation for terminal physiology experiments P55-68

The ferret was sedated with ketamine (20 mg kg$^{-1}$ im). Atropine (0.16–0.8 mg kg$^{-1}$ im) and dexamethasone (0.5 mg kg$^{-1}$ im) were administered to reduce bronchial and salivary secretion and to reduce inflammation, respectively. The animal was next anesthetized with a mixture of isoflurane, oxygen, and nitrous oxide through a mask, and a tracheostomy was performed. The animal was then ventilated with 1.5–3% isoflurane in a 2:1 mixture of nitrous oxide and oxygen. A cannula was inserted into the intraperitoneal (ip) cavity for delivery of neuromuscular blockers and Ringer solution (3 ml kg$^{-1}$ hr$^{-1}$), and the animal was inserted in a custom stereotaxic frame that did not obstruct vision. All wound margins were infused with bupivacaine. A suture was made in the scalp and the skin resected over the posterior half of the skull. A small opening in the cranium was drilled over V1 with a dental drill (Medidenta). The dura covering the brain at the craniotomy site was removed over a 1 mm × 1 mm area to allow for electrode placement. Craniotomies were made over V1 of each hemisphere. The eyelids were sutured open, and contact lenses were placed on the eyes to prevent corneal damage. Before visual stimulation commenced, the ferret was paralyzed with the neuromuscular blocker gallamine triethiodide (10–30 mg kg$^{-1}$ hr$^{-1}$) through the ip cannula to suppress spontaneous eye movements, and the nitrous oxide : oxygen mixture was adjusted to 1:1. The animal's ECG was continuously monitored to ensure adequate anesthesia, and the percentage of isoflurane was increased if the ECG indicated any distress. Body temperature was maintained at 37°C. At the conclusion of the experiment, the animal was killed with an overdose of sodium pentobarbitol (200 mg/kg, IP) and transcardially perfused to retrieve the brain for histology.

## Electrophysiological recordings

32 channel electrodes (Plexon S probes, 50 um inter-tetrode spacing; 50 µm intra-tetrode spacing) were used for all recordings. The signal was amplified using the RHD2000 amplifying/digitizing chip and USB interface board (Intan Technologies). Stimulus timing information was acquired using a Micro1401 acquisition board and Spike2 software (Cambridge Electronic Design). Spike sorting was performed offline using JRClust running in Matlab (*Jun et al., 2017*). An electrode was inserted into the brain using a Sutter Instruments MP-285 manipulator. To reduce sampling bias, we recorded from any site that had a signal-to-noise ratio sufficient for isolation and had a response that appeared to be modulated by the presentation of drifting gratings. Data are reported from all units that are responsive enough to be included in analysis (see below). After finishing the recording at one site, the electrode was lowered at least 640 microns before attempting to identify a suitable subsequent recording site. The experiment was concluded when successful recordings were made from one or both hemispheres.

## Locating monocular neurons

After placing the electrode at a minimum depth of 200 µm in the brain, entering the brain at an angle of 30–45 degrees, the search for monocular neurons commenced. The electrode was driven into the

brain until monocular neurons were discovered. A population of neurons was deemed monocular if their receptive fields were in the periphery of the eye contralateral to the recording hemisphere. Monocularity was ensured by determining whether movement in the central visual and ipsilateral visual fields evoked a response; any such responses indicated binocularity. If no purely monocular cells were found in a given penetration, the electrode was removed and another penetration was made elsewhere in the brain.

## Locating the optic disk

An indirect ophthalmoscope (Heine Omega 600) and handheld 78 or 90D lens (Volk) were used to locate the optic disk and determine the orientation of the eye. This was necessary as the orientation of the eye is necessary for determining the true eccentricity of receptive fields. When one investigator had the optic disk in the center of view of the indirect ophthalmoscope, another investigator introduced a wooden rod into the line of sight and moved it until the end of the rod was centered on the image of the optic disk (also the center of view). The position of the end of the rod was then measured in X, Y, and Z relative to the animal's eye to make the optic disk direction.

## Visual stimuli

Visual stimuli were created in MATLAB (MathWorks) using the Psychophysics Toolbox (*Brainard, 1997*; *Pelli, 1997*) and displayed on a 21-inch flat face CRT monitor (GDM-520, Sony) with a resolution of 800×600 and a refresh rate of 100 Hz. We manually mapped receptive fields by displaying circular patches of drifting sinusoidal gratings at different positions and moving the monitor to accommodate different eccentricities while listening to the responses on a loudspeaker.

Drifting grating stimuli were full-field, drifting sinusoidal gratings (4 s duration; 3.5 s interstimulus interval) presented pseudorandomly, with direction of motion (in steps of 45°) in either of the two directions orthogonal to the axis of orientation. Each individual grating stimulus was full screen and had a single set of parameters (direction, spatial frequency, temporal frequency) and was separated from the other stimuli by a gray screen interstimulus interval. We ran two sets of stimuli where multiple grating parameters were covaried in order to sample a wide variety of receptive field properties in our multi-neuron recordings. In the first set, orientation (angles 0°/horizontal, 45°, 90°/vertical, 135°) was co-varied with 8 different spatial frequencies (0.04, 0.08, 0.16, 0.24, 0.32, 0.64, 0.90, and 1.25), and 6 different contrasts (0.04, 0.08, 0.16, 0.32, 0.64, and 1), while temporal frequency was held constant at 4 Hz. In this first set of stimuli, gratings drifted back and forth every 4 cycles; for example, horizontal stimuli drifted upward and then downward. In a second set of stimuli, direction (0°/up, 45°, 90°/right, 135°, 180°/down, 225°, 270°/left, 315°) was co-varied with stimuli of 7 different temporal frequencies (0.5, 1, 2, 4, 8, 16, and 32 Hz) at 100% contrast and a fixed spatial frequency of 0.1 cycles/° visual angle. In this second set of stimuli, gratings drifted in a single direction and did not move back and forth. Each stimulus set had a blank, control stimulus that was the same duration as the other stimuli where the screen did not change. The first set of stimuli was repeated five times, and the second set seven times.

## Data analysis

Responses were determined by examining either the mean response during the entire stimulus or by taking the F1 component of the response. The same analysis (mean, or F1 at the temporal frequency of the stimulus of interest) was performed for the closest blank (control) stimulus and the response to the control stimulus was subtracted to yield the stimulus-driven response. For each tuning curve for orientation, direction, and spatial frequency analyses, we examined whether the mean or the F1 response was higher, and used those responses for analysis and fits. For temporal frequency analysis, we only used mean responses because we found a profound response suppression in many cells. Inclusion/exclusion: For each stimulus type, we examined the set of all responses to visual stimuli and blanks with an ANOVA test to evaluate the null hypothesis that the mean response to all of these stimuli was the same; cells with a $p<0.05$ to this visual responsiveness test were included in fits and analyses, and cells with $p>0.05$ were excluded.

Orientation selectivity was examined in the first set of stimuli at the preferred spatial frequency for each cell. Circular variance was calculated in orientation space (*Ringach et al., 2002*; *Mazurek et al., 2014*) with the equation V=1 - |R |, where R is the resultant calculated as:

$$R = \frac{\sum_k r_k e^{i20_k}}{\sum_k r_k}$$

where $r_k$ was the mean spike rate in response to a grating drifting with angle $\theta_k$.

Tuning curves from cells that showed significant variation across orientation stimuli and blank ANOVA test (*Mazurek et al., 2014*) were fit with a double Gaussian to obtain the angle preference.

Direction selectivity was examined in the second set of stimuli for the preferred temporal frequency of each cell (unless noted). Circular variance was calculated in direction space with the following equation:

$$1 - DirCirVar = \left| \frac{\sum_k R(\theta_k)\exp(i\theta_k)}{\sum_k R(\theta_k)} \right|$$

Preferred spatial frequency was determined by responses to the first stimulus battery at the preferred orientation at 100% contrast. Preferred temporal frequency was determined by responses to the second stimulus battery at the preferred direction. Each was fitted with a model due to *Movshon et al., 2005*:

$$R(f) = k * \exp\left(-\left(\frac{f}{fc}\right)^2\right) \bigg/ \left[1 + \left(\frac{fh}{f}\right)^\beta\right]$$

Low-frequency cutoffs were determined by following the fit from its peak to lower frequencies until the first frequency where the response dropped to half its maximum (L50); high-frequency cutoffs (H50) were determined by the same procedure, except that the fit was followed from the peak towards higher frequencies (*Moore et al., 2005*; *Heimel et al., 2005*). If the response never dropped below half its maximum, then the low-frequency cutoff was said to be 0 or the high-frequency cutoff was said to be infinity. Temporal frequency tuning bandwidth was defined as log2(H50/L50).

We defined a low-pass index as the response to the lowest temporal frequency tested (in this case 0.5 Hz) to the maximum response obtained to the set of temporal frequencies shown. LPI = $\frac{R(TF=0.5Hz)}{max(R(TF=0.5Hz),R(TF=1Hz),...,R(TF=32Hz))}$ . If a cell exhibited the highest firing for a temporal frequency of 0.5 Hz then it would have a low-pass index of 1. If it exhibited a similar firing rate in response to a temporal frequency of 0.5 Hz even if the preferred temporal frequency were higher, then the low-pass index would still be near 1. If the cell responded poorly at a temporal frequency of 0.5 Hz, then it would have a low-pass index near 0.

Contrast responses were fit with Naka-Rushton functions as described in *Peirce, 2007* and sensitivity determined by the reciprocal of the contrast for which the response exceeds 5 standard deviations of the response to a blank screen. The Naka-Rushton function is defined as:

$$R = R_{max}\frac{C^n}{C_{50}^n + C^n} + B$$

where $R_{max}$ is the maximum response of the neuron, $c_{50}$ is the contrast at which the response is halfway between baseline and maximum, c is the contrast of the stimulus presented to the neuron, and B is the baseline response of the neuron. n is a factor simply referred to as the exponent.

## Statistics
Index values were examined with linear mixed effects models in Matlab (function: fitlme), where treatment condition (control, EO1contra, EO1ipsi, EO2) were fixed effects and animal identities were random effects.

## Immunohistochemistry

Upon completion of experiments, an electrode coated in the fluorescent dye DiI (*DiCarlo et al., 1996*) was inserted at the location and depth and left in place for 20 min. Animals were then transcardially perfused and the brain was placed in 4% paraformaldehyde in 0.1 M PBS at 4 °C for 24 hr and then moved to 10% sucrose in PBS for 24–48 hr. This was followed by placement in 30% sucrose in PBS at 4 °C until sectioning. The brain was sectioned sagittally into 100 µm sections using a sliding microtome (Leica SM2010R). We washed sections in 0.1 M PBS 3×5 min and permeabilized in 0.3% Triton-X 100 diluted in PBS for 2 hr at room temperature on a shaker. Then slices were incubated in fluorophore-conjugated anti-NeuN antibody (Alexa Fluor 488 Rabbit anti NeuN, Millipore ABN78A4) at a 1:300 dilution overnight (>12 hr) at room temperature on a shaker. Sections were then washed 3×5 min in PBS and mounted on slides and allowed to air dry. Slides were then cover-slipped with Fluoromount-G media (Electron Microscopy Sciences, Ft. Washington, PA) and edges were sealed using nail polish. Histological sections were viewed using a fluorescent microscope (Keyence BX-Z 710) and electrode tracks were reconstructed using DiI dye traces.

## Acknowledgements

This work was funded by NIH EY022122 (SDV). We thank Jiwon Sun for help with histology and some experiments, Audrey Jordan and Rommy Sierra for some figure editing, and members of the Van Hooser lab for comments.

## Additional information

### Funding

| Funder | Grant reference number | Author |
| --- | --- | --- |
| National Eye Institute | EY022122 | Stephen D Van Hooser |

The funders had no role in study design, data collection and interpretation, or the decision to submit the work for publication.

### Author contributions

Sophie V Griswold, Conceptualization, Formal analysis, Investigation, Visualization, Methodology, Writing – original draft, Writing – review and editing; Stephen D Van Hooser, Conceptualization, Data curation, Software, Formal analysis, Supervision, Funding acquisition, Validation, Visualization, Project administration, Writing – review and editing

### Author ORCIDs

Sophie V Griswold ⓘ https://orcid.org/0000-0003-2340-9602
Stephen D Van Hooser ⓘ https://orcid.org/0000-0002-1112-5832

### Ethics

This study was performed in strict accordance with the recommendations in the Guide for the Care and Use of Laboratory Animals of the National Institutes of Health. All of the animals were handled according to approved institutional animal care and use committee (IACUC) at Brandeis University (16003, 19010, 22010-A).

Reviewer #1 (Public review): https://doi.org/10.7554/eLife.106513.3.sa1
Reviewer #2 (Public review): https://doi.org/10.7554/eLife.106513.3.sa2
Author response https://doi.org/10.7554/eLife.106513.3.sa3

## Additional files

### Supplementary files

MDAR checklist

## Data availability

Data were managed with the Neuroscience Data Interface (*García Murillo et al., 2022*) and shared with NDI Cloud at https://doi.org/10.63884/ndic.2025.28xb47y1 (*Griswold and Van Hooser, 2025*).

The following dataset was generated:

| Author(s) | Year | Dataset title | Dataset URL | Database and Identifier |
|---|---|---|---|---|
| Griswold SV, Van Hooser SD | 2025 | Premature vision drives aberrant development of response properties in primary visual cortex | https://doi.org/10.63884/ndic.2025.28xb47y1 | NDI Cloud, 10.63884/ndic.2025.28xb47y1 |

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
