## [Editor Report · eLife Assessment]

This carefully conducted study aims to understand how the early visual experience of premature infants induces lasting deficits, including compromised motion processing. The authors address this **important** question in a ferret animal model, exposing the developing visual system prematurely to patterned visual input by opening one or both eyes at a time when both retinal waves and light traveling through closed lids can drive sensory responses. **Convincing** evidence is presented, suggesting that eye opening at this time impacts temporal frequency tuning and elevates spontaneous firing rates. These findings will have great relevance for neuroscientists studying visual system development, particularly in the context of premature birth.

---

## [Referee Report · Reviewer #1 (Public review)]

The authors note that very premature infants experience the visual world early and, as a consequence, sustain lasting deficits including compromised motion processing. Here they investigate the effects of early eye opening in ferret, choosing a time point after birth when both retinal waves and light traveling through closed lids drive sensory responses. The laboratory has long experience in quantitative studies of visual response properties across development and this study reflects their expertise.

The investigators find little or no difference in mean orientation and direction selectivity, or in spatial frequency tuning, as a result of early eye opening but marked differences in temporal frequency tuning. These changes are especially interesting as they relate to deficits seen in prematurely delivered children. Temporal frequency bandwidth for responses evoked from early-opened contralateral eyes were broader than for controls; this is the case for animals in which either one or both eyes were opened prematurely. Further, when only one eye was opened early, responses to low temporal frequencies were relatively stronger.

The investigators also found changes in firing rate and sign of response to visual stimuli. Premature eye-opening increased spontaneous rates in all test configurations. When only one eye was opened early, firing rates recorded from the ipsilateral cortex were strongly suppressed, with more modest effects in other test cases.

As the authors' discussion notes, these observations are just a starting point for studies underlying mechanism. The experiments are so difficult to perform and so carefully described that the results will be foundational for future studies of how premature birth influences cortical development.

---

## [Referee Report · Reviewer #2 (Public review)]

In this paper, Griswold and Van Hooser investigate what happens if animals are exposed to patterned visual experience too early, before its natural onset. To this end, they make use of the benefits of the ferret as a well-established animal model for visual development. Ferrets naturally open their eyes around postnatal day 30; here, Griswold and Van Hooser opened either one or both eyes prematurely. Subsequent recordings in the mature primary visual cortex show that while some tuning properties like orientation and direction selectivity developed normally, the premature visual exposure triggered changes in temporal frequency tuning and overall firing rates. These changes were widespread, in that they occurred even for neurons responding to the eye that was not opened prematurely. These results demonstrate that the nature of the visual input well before eye opening can have profound consequences on the developing visual system.

The conclusions of this paper are well supported by the data, but in the initially submitted version of the paper, there were a few questions regarding the data processing and suggestions for the discussion:

(1) The assessment of the tuning properties is based on fits to the data. Presumably, neurons for which the fits were poor were excluded? It would be useful to know what the criteria were, how many neurons were excluded, and whether there was a significant difference between the groups in the numbers of neurons excluded (which could further point to differences between the groups).

(2) For the temporal frequency data, low- and high-frequency cut-offs are defined, but then only used for the computation of the bandwidth. Given that the responses to low temporal frequencies change profoundly with premature eye opening, it would be useful to directly compare the low- and high-frequency cut-offs between groups, in addition to the index that is currently used.

(3) In addition to the tuning functions and firing rates that have been analyzed so far, are there any differences in the temporal profiles of neural responses between the groups (sustained versus transient responses, rates of adaptation, latency)? If the temporal dynamics of the responses are altered significantly, that could be part of an explanation for the altered temporal tuning.

(4) It would be beneficial for the general interpretation of the results to extend the discussion. First, it would be useful to provide a more detailed discussion of what type of visual information might make it through the closed eyelids (the natural state), in contrast to the structured information available through open eyes. Second, it would be useful to highlight more clearly that these data were collected in peripheral V1 by discussing what might be expected in binocular, more central V1 regions. Third, it would be interesting to discuss the observed changes in firing rates in the context of the development of inhibitory neurons in V1 (which still undergo significant changes through the time period of premature visual experience chosen here).

---

## [Author Response]

The following is the authors’ response to the original reviews.

**Reviewer #1 (Public Review):**
(1) Figure 1: It might be simpler to streamline acronyms for different test cases, e.g, E01contra, E01 ipsi (rather than EO1IPS), E02, and control. Thus, it would be possible to label each of the three schematic panels as E01, E02, control.Please describe what the dots in the brain mean and move the V1 label so it does not occlude dots.Please make clear that the "track reconstructions" are the bright spheres in the micrographs (there are track-like elements in some micrographs which may be tears or?)

Thank you. We relabeled the groups as control, EO1contra, EO1ipsi, and EO2. These were changed in all figures and in the document at several places.

We indicated in the new caption that “Dots schematize ocular dominance columns”.

We indicated that electrode track penetrations were the “(bright spots at right/posterior)”.

(2) Figure 2: Should "horizontal" be vertical (line 556) of the caption? When describing the scale bar for firing rate, please explain the meaning of italicized vs regular font.Please make the purple lines in Figures I and J easier to see (invisible in my PDF).Not quite clear what is significantly different from what when viewing the figure at a glance. Would it be possible to clarify using standard methods?

Yes, it should say vertical, thank you. We explained the italics (they denote the standard scale bar size if no number is provided.)

We changed the purple lines to yellow in all figures.

We added comparison bars that help indicate significance.

(3) Figures 3-5. Please make corrections like those noted above.

Yes, we applied the previous changes to Figures 3 - 5.

(4) Minor. Sometimes the authors spell out temporal frequency and sometimes abbreviate it. Perhaps adopt a consistent style.

Fixed, thanks.

**Reviewer #2 (Public Review):**
(1) The assessment of the tuning properties is based on fits to the data. Presumably, neurons for which the fits were poor were excluded? It would be useful to know what the criteria were, how many neurons were excluded, and whether there was a significant difference between the groups in the numbers of neurons excluded (which could further point to differences between the groups).

Yes, this is an important omission, thank you for catching it. We now write in methods (line 213): “ Inclusion/exclusion: For each stimulus type, we examined the set of all responses to visual stimuli and blanks with an ANOVA test to evaluate the null hypothesis that the mean response to all of these stimuli were the same; cells with a p<0.05 to this visual responsiveness test were included in fits and analyses, and cells with p>0.05 were excluded. ”

(2) For the temporal frequency data, low- and high-frequency cut-offs are defined, but then only used for the computation of the bandwidth. Given that the responses to low temporal frequencies change profoundly with premature eye opening, it would be useful to directly compare the low- and high-frequency cut-offs between groups, in addition to the index that is currently used.

We now provide this data in Figure 3 - figure supplement 1 **.**

(3) In addition to the tuning functions and firing rates that have been analyzed so far, are there any differences in the temporal profiles of neural responses between the groups (sustained versus transient responses, rates of adaptation, latency)? If the temporal dynamics of the responses are altered significantly, that could be part of an explanation for the altered temporal tuning.

This is a great topic for future studies. Unfortunately, with drifting gratings, it is difficult to establish these properties, which could be better assessed with standing or square-wave-modulated gratings or other stimuli. We did not run standing gratings in our battery of stimuli for this initial study.

(4) It would be beneficial for the general interpretation of the results to extend the discussion. First, it would be useful to provide a more detailed discussion of what type of visual information might make it through the closed eyelids (the natural state), in contrast to the structured information available through open eyes. Second, it would be useful to highlight more clearly that these data were collected in peripheral V1 by discussing what might be expected in binocular, more central V1 regions. Third, it would be interesting to discuss the observed changes in firing rates in the context of the development of inhibitory neurons in V1 (which still undergo significant changes through the time period of premature visual experience chosen here).

Thank you, good ideas. Let’s take these three suggestions in turn.

First, in the discussion, we added a subsection “ Biology of early development in mustelids ” that focuses on the developmental conditions of wild and laboratory animals:

In the wild, mustelids raise their young in nests in the ground, in cavities such as holes in trees or caves, or in areas of dense vegetation (Ruggiero et al. 1994). They may move the young from one nest to another as they grow, but otherwise the young are primarily in the relatively dark nest. It is highly likely that some light penetrates and that information about the 24-hour cycle is available, but the light is likely to be dim and unlikely to provide a basis for high luminance, high contrast stimulation through the closed lids. The animals begin to spend substantial time outside the nest after eye opening.

The ferret is a domesticated strain of the European polecat. In laboratory settings, ferret jills give birth and keep their kits in a nest box. A laboratory typically maintains a 24-hour cycle with 12 or 14 hours of light, and the light reaching the closed lids must first pass through the cage, the nest box, and the nesting material. Therefore, developing ferrets have an obvious circadian light signal but the light available for image formation is likely dim and of low contrast.

Although the light that reaches the close lids in developing ferrets is likely to be relatively dim, and any image-forming signal passing through the closed lids would be highly filtered in luminance, spatial frequency, and contrast, it is important to remember that visual input before natural eye opening (through the closed lids) can drive activity in retina, LGN, and cortex (Huttenlocher 1967, Chapman and Stryker 1993, Krug et al., 2001, Akerman et al., 2002,Akerman et al., 2004). Further, orientation selectivity can be observed through the closed lids (Krug et al., 2001), indicating that some coarse image-forming information does make it through the closed lids.

Second, we added text speculating about binocular cortex (lines 492 - 500): … our recordings were performed in monocular cortex so that we could be sure of the developmental condition of the eye that drove the classic responses. It is interesting to speculate about what might occur more centrally in binocular visual cortex. Ocular dominance shifts are not induced when one eye is opened prematurely (Issa et al 1999), indicating that ocular dominance plasticity is not engaged at this early stage, but one might imagine that the impacts on temporal frequency and spontaneous firing rates would still be present.

Third, on inhibition, we added a paragraph (lines 502 - 509):

We introduced premature patterned vision at a time when cortical inhibition is undergoing substantial changes. GABAergic signaling has already undergone its switch (Ben-Ari, 2002) from providing primarily depolarizing input to hyperpolarizing input by P21-23 (Mulholland et al., 2021). In the days prior to eye opening, inhibitory cells exhibit activity that is closely associated with the emerging functional modules that will reflect orientation columns (Mulholland et al., 2021), but do not yet exhibit selectivity to orientation, in contrast to excitatory neurons, which do exhibit selectivity to orientation at that time (Chang and Fitzpatrick, 2022).

(5) In the methods section, the statement 'actively kept in nesting box' is unclear. Presumably this means that the jill prevents the kits from leaving the nesting box? It also would be worth at least mentioning in this context that there obviously are still visual events in the nesting box too.

Thanks. We improved this description (lines 118 - 121): Ferret kits in laboratory housing receive limited visual stimulation through their closed lids, as the mother actively keeps the kits in their relatively dark nest . In order to ensure that animals with early-opened eyes actually had patterned visual experience (and animals with closed lids had the same stimulation filtered through the lids) , animals were brought to the lab for 2 hours a day for 4 consecutive days beginning at P25.

(6) The stimulus presentation could be more clearly described. Is every stimulus presented in an individual trial (surrounded by periods with a blank screen), or are all stimuli shown as a continuous sequence? The description of the parameter screening is also potentially confusing ('orientation was co-varied with stimuli consisting of drifting gratings at different spatial frequencies' sounds as if there are separate stimuli for orientation; might be better to say something like 'in the first set, orientation, spatial frequency, ... were covaried...')

Yes, thank you, we fixed this (lines 184 - 201). We deleted the text indicated and added a sentence “Each individual grating stimulus was full screen and had a single set of parameters (direction, spatial frequency, temporal frequency), and was separated from the other stimuli by a gray screen interstimulus interval.”. We also deleted a repetition of 100% contrast in the description of the second set.

(7) Description of low-pass index is unclear. What is the 'largest temporal frequency response observed'? The maximum response or the response to the largest temporal frequency tested?

Thanks. We added a paragraph at line 236:

We defined a low-pass index as the response to the lowest temporal frequency tested (in this case 0.5 Hz) to the maximum response obtained to the set of temporal frequencies shown. LPI = R(TF=0.5 Hz)/max(R(TF=0.5Hz), R(TF=1Hz), … R(TF=32Hz)). If a cell exhibited the highest firing for a temporal frequency of 0.5 Hz, then it would have an low-pass index of 1. If it exhibited a similar firing rate in response to a temporal frequency of 0.5 Hz even if the preferred temporal frequency were higher, then the low-pass index would still be near 1. If the cell responded poorly at a temporal frequency of 0.5 Hz, then it would have a low-pass index near 0.

(8) The discussion should also cite the results of strobe-reared cats by Pasternak et al (1981 and 1985).

Thank you for pointing out the omission. We now write (lines 430-435): Cats raised in a strobe-light environment (mostly after eye opening) exhibited strong changes in subsequent direction selectivity (Kennedy and Orban 1983; Humphrey and Saul 1998) and behavioral sensitivity to motion (Pasternak et al., 1981; Pasternak et al., 1985) that partially recovers with motion detection training . However, temporal frequency tuning of these animals has not been reported in detail. Pasternak et al (1981) reported that strobe-reared cats exhibited greater difficulty in distinguishing slow moving stimuli from static stimuli compared to controls, an ability that slightly improved with practice, suggesting possible temporal frequency deficits.

(9) Finally, it would be useful to include a mention of the early development of MT in marmosets in the discussion of impacts of prematurity on motion vision (Bourne & Rosa 2006).

Yes, thank you. We cited Bourne & Rosa and also Lempel and Nielsen (for ferret PSS). (Lines 492-501):

Several other basic mechanistic questions remain unanswered. It is unclear where in the visual circuit cascade these deficits first arise. Does the lateral geniculate nucleus or retina exhibit altered temporal frequency tuning? Is the influence of the patterned visual stimulation instructive, so that if one provided premature stimulation with only certain temporal frequencies, one would see selectivity for those temporal frequencies, or would tuning always be broad? Other questions remain concerning the top-down influence on V1 from “higher” motion areas such as MT (monkeys) or PSS (ferret); MT exhibits mature neural markers earlier than V1 (Bourne and Rosa, 2006), and suppression of PSS impacts motion selectivity in V1 (Lempel and Nielsen, 2021). Future studies will be needed to address these questions.